# CAUSAL INFERENCE FOR KNOWLEDGE GRAPH COMPLETION

## ABSTRACT

The basis of existing knowledge graph completion (KGC) models is to learn the correlations in data, such as the correlation between entities, relations and scores of triplets. Since correlation is not as reliable as causation, correlation-driven KGC models are weak in interpretability and suffer from the data bias issues. In this paper, we propose causal KGC models to alleviate the data bias issues by leveraging causal inference framework. Our method is intuitive and interpretable by utilizing causal graphs, controllable by using intervention techniques and model-agnostic. Causal graphs allow us to explain the causal relationships between variables and study the data generation processes. Under the causal graph, data bias can be seen as confounders. Then we block the bad effect of confounders by intervention operators to mitigate the data bias issues. Due to the difficulty of obtaining randomized data, causal KGC models pose unique challenges for evaluation. Thus, we show a method that makes evaluation feasible. Finally, we show a group theory view for KGC, which is equivalent to the view of causal but further reveals the causal relationships. Experimental results show that our causal KGC models achieve better performance than traditional KGC models on three benchmark datasets.

## 1 INTRODUCTION

A knowledge graph (KG) consists of a large number of triplets in the form of *(head entity, relation, tail entity)*. Many KGs suffer from the incompleteness problem. To complement the KGs, knowledge graph completion (KGC) models define a scoring function to measure the likelihood of triplets. The core of traditional KGC models is to learn the correlation in data, such as the correlation between entities or relations and scores of triplets. Since correlation is not as reliable as causation, purely modeling the correlation leads to poor interpretability and the data bias issues. For example, due to ignoring popularity bias in KG data, KGC models are biased towards popular entities and relations (Mohamed et al., 2020).

In this paper, we propose causal KGC models to solve the data bias issues by utilizing causal inference techniques (Pearl, 2009b). Our method is model-agnostic and just needs to add an extra term to the traditional KGC models. Causal inference defines causal graphs to describe the causal relationships between variables. Causal graphs can help build intuitive, interpretable and controllable KGC models. Traditional KGC models are only concerned with the correlations in the data, while ignoring the causation and the data generation process, which can lead to incorrect correlations between entities, relations and scores of triplets. Causal graphs allow us to explain the causal relationships between variables and study the data generation processes. Under the causal graph, data bias can be seen as confounders, where confounders in KG data are variables that simultaneously affect entities or relations and scores of triplets. We utilize intervention operators to eliminate the bad effect of confounders, which remove the path from confounders to entities and relations in the causal graph. Then we can estimate the causal effect or correct correlations in KG data by backdoor adjustment formula (Pearl, 2009b).

Causal KGC models present special challenges for evaluation, which need to evaluate on a randomized test set. However, a randomized test set is often difficult or infeasible to obtain. Therefore, we define a new evaluation metric to measure the performance of causal KGC models based on the popularity of entities and relations.

The main feature of causal is invariance or symmetry (Arjovsky et al., 2020). Group theory is a language to describe symmetry. Thus, we finally show a view of group theory for KGC, which is equivalent to the view of causal but further uncovers the causal relationships. The view of group theory transcends the view of causal and shows potential applications.

The main contributions of this paper are listed below:

1. To the best of our knowledge, we are the first to show the necessity of introducing causation into KGC and apply causal inference to KGC.

2. We propose causal KGC models to enhance the interpretability of KGC models and alleviate the data bias issues. Then we show a method to evaluate causal KGC models on observation datasets.

3. We show a view of group theory for KGC to further reveal the causal relationships.

4. We empirically show that causal KGC models outperform traditional KGC models on three benchmark datasets.

## 2 BACKGROUND

In this section, we introduce the related background of our model, knowledge graph completion and causal inference.

### 2.1 KNOWLEDGE GRAPH COMPLETION

Let $\mathcal{E}$ denote the set of entities and $\mathcal{R}$ denote the set of relations, a KG is composed of a set of triplets $\mathcal{D} = \{(h, r, t)\} \subset \mathcal{E} \times \mathcal{R} \times \mathcal{E}$, where $h$ is a head entity, $r$ is a relation and $t$ is a tail entity. Lacroix et al. (2018) propose to augment every triplet $(h, r, t)$ in $\mathcal{D}$ with its inverse triplet $(t, r^{-1}, h)$. With this augmentation, KGC can be formulated as predicting the tail entities that satisfy a query $(h, r, ?)$. A KG can also be represented by a 3rd-order binary tensor $\boldsymbol{X} \in \{0, 1\}^{|\mathcal{E}| \times |\mathcal{R}| \times |\mathcal{E}|}$ with $\boldsymbol{X}_{h,r,t} = 1$ if $(h, r, t) \in \mathcal{D}$ and $\boldsymbol{X}_{h,r,t} = 0$ if $(h, r, t) \notin \mathcal{D}$.

KGC models define a scoring function $f(h, r, t)$ to measure the likelihood of a triplet $(h, r, t)$ based on the corresponding embedding $(\boldsymbol{h}, \boldsymbol{r}, \boldsymbol{t})$. A number of KGC models have been proposed (Zhang et al., 2021a), we list four popular KGC models that we consider in our experiments.

TransE (Bordes et al., 2013), a representative model of translation-based models, defines the scoring function as the negative distance between $\boldsymbol{h} + \boldsymbol{r}$ and $\boldsymbol{t}$, i.e.,

$$f(h, r, t) = -\|\boldsymbol{h} + \boldsymbol{r} - \boldsymbol{t}\|$$

where $(\boldsymbol{h}, \boldsymbol{r}, \boldsymbol{t})$ is the corresponding embeddings of $(h, r, t)$, $\boldsymbol{h}, \boldsymbol{r}, \boldsymbol{t} \in \mathbb{R}^n$, $n$ is the dimension of embedding and $\|\cdot\|$ is a norm of a vector. RotatE (Sun et al., 2018) generalizes the embedding from real vector space to complex vector space to model various relation patterns, and the scoring function is defined as

$$f(h, r, t) = -\|\boldsymbol{h} \odot \boldsymbol{r} - \boldsymbol{t}\|$$

where $\boldsymbol{h}, \boldsymbol{r}, \boldsymbol{t} \in \mathbb{C}^n$ and $\odot$ is Hadamard product.

DistMult (Yang et al., 2014), a representative model of multiplicative models, defines the scoring function as the inner product of $\boldsymbol{h}$, $\boldsymbol{r}$ and $\boldsymbol{t}$, i.e.,

$$f(h, r, t) = \sum_{i=1}^{n} \boldsymbol{h}_i \boldsymbol{r}_i \boldsymbol{t}_i$$

where $\boldsymbol{h}, \boldsymbol{r}, \boldsymbol{t} \in \mathbb{R}^n$. ComplEx (Trouillon et al., 2017) extends DistMult to complex vector space to handle asymmetric relation patterns and defines the scoring function as

$$f(h, r, t) = \text{Re}(\sum_{i=1}^{n} \boldsymbol{h}_i \boldsymbol{r}_i \boldsymbol{t}_i^*)$$

where $\boldsymbol{h}, \boldsymbol{r}, \boldsymbol{t} \in \mathbb{C}^n$, $\boldsymbol{t}_i^*$ is the complex conjugate of $\boldsymbol{t}_i$ and $\text{Re}(\cdot)$ is the real part of a complex number.

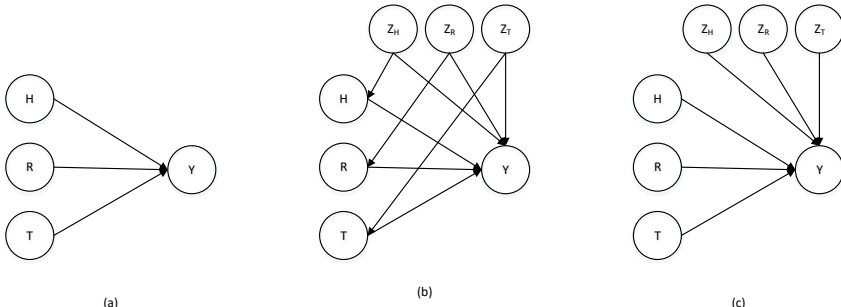

Figure 1: (a) describes the causal graph of traditional KGC models. (b) describes the causal graph with three confounders. (c) describes the causal graph after the intervention $do(H, R, T)$.

## 2.2 CAUSAL INFERENCE

Causal inference is the process of inferring the causal relationships from data (Yao et al., 2021). There are two representative frameworks for causal inference: structural causal models (SCMs) proposed by Pearl (2009b) and potential outcome framework developed by Rubin (1974). As shown in (Pearl, 2009a), the two frameworks are logically equivalent. Since the causal graph in SCMs describes the causal relationships more intuitively, we select the SCMs framework in this paper.

SCMs abstract the causal relationships between variables into a set of functions and then estimate the causal effects of an intervention or a counterfactual. Every SCM is associated with a causal graph, which is a directed acyclic graph where the nodes denote variables and the edges indicate causal relationships between variables.

Given the causal graph, a fundamental manipulation on the causal graph is the intervention. Technically, the intervention on a variable $H$ is formulated with $do$-calculus, $do(H = h)$, which blocks the effect of $H$'s parents and sets the value of $H$ as $h$. For example, $do(H = h)$ in Figure 1(b) will remove the path $Z_H \rightarrow H$ and force $H$ to be $h$. In SCMs framework, the probability function $P(Y|do(H))$ describes the causal effect of a variable $H$ on a variable $Y$.

The backdoor adjustment formula is commonly used to estimate the causal effect $P(Y|do(H))$. Given a causal graph in which a set of variables $Z_H$ are the parents of a variable $H$, then the causal effect $P(Y|do(H))$ can be obtained by the backdoor adjustment formula: $P(Y = y|do(H = h)) = \sum_{z_h \in Z_H} P(y|h, z_h)P(z_h)$. Thus, the causal effect $P(Y = y|do(H = h))$ is the weighted sum of the conditional probability $P(y|h, z_h)$. Then we can estimate $P(y|h, z_h)$ from the observation data to obtain $P(Y = y|do(H = h))$.

## 3 METHOD

In this section, we first propose our causal KGC models by utilizing causal inference techniques. Then, we show the method of evaluating causal KGC models. Finally, we show a group theory view for KGC to further reveal the causal relationships.

### 3.1 CAUSAL KGC MODELS

**Data Bias in KGs** Data bias in KGs refers to the biased data collection that does not faithfully reflect the likelihood of triplets. Many popular KGs (e.g. DBpedia, Wikidata, and YAGO) are automatically constructed from unstructured text by using information extraction algorithms (Ji et al., 2021). The collected KG data often suffers from data bias, such as popularity bias, algorithm bias and so on. For example, Mohamed et al. (2020) show that the distribution of entities and relations in the benchmark KGs is highly skewed. The collected KG data is also affected by the information extraction algorithms, which may only extract simple knowledge from the unstructured text while

ignoring complex knowledge. Existing correlation-driven KGC models not only learn the desired likelihood of triplets but also the data bias, which leads to incorrect correlation.

**A Causal View for KGC**   Causal inference allows us to find the fundamental cause of data bias by studying the generation process of KG data and alleviates the effect of data bias. In most cases, data bias can be seen as confounders in a causal graph, where confounders in KG data are variables that simultaneously affect entities or relations and scores of triplets. Ignoring confounders can lead to incorrect correlations between entities, relations and scores of triplets. To eliminate confounding effects, we abstract the data generation process into a causal graph, identify confounders, and then block the effect of confounders by intervention.

Causal graphs are intuitive and allow us to explain the causal relationships between variables. Figure 1 shows three causal graphs. Figure 1(a) describes the causal graph of traditional KGC models, which ignores the confounders. Figure 1(b) describes the causal graph with three confounders. Figure 1(c) describes the causal graph after the intervention of the causal graph in Figure 1(b). The nodes and edges are illustrated as follows:

1. **Nodes**: Node $H$ denotes the head entity variable. Node $R$ denotes the relation variable. Node $T$ denotes the tail entity variable. Node $Y$ denotes the score of triplets variable, which measures the likelihood of triplets. Node $Z_H$, $Z_R$ and $Z_T$ denote variables that are not explicitly considered by traditional KGC models, e.g., the popularity of entities or relations.

2. **Edges**: Edges $\{H, R, T\} \rightarrow Y$ denote that $\{H, R, T\}$ are the cause of $Y$, which is exactly what the traditional KGC models consider. Edges $\{H, R, T, Z_H, Z_R, Z_T\} \rightarrow Y$ denote that $\{H, R, T, Z_H, Z_T, Z_T\}$ are the cause of $Y$, which have three extra edges $Z_H \rightarrow Y$, $Z_R \rightarrow Y$ and $Z_T \rightarrow Y$ compared to edges $\{H, R, T\} \rightarrow Y$. Edge $Z_H \rightarrow Y/Z_R \rightarrow Y/Z_H \rightarrow Y$ means that $Z_H/Z_R/Z_T$ is also contributed to $Y$. Edge $Z_H \rightarrow \{H\}/Z_R \rightarrow \{R\}/Z_T \rightarrow \{T\}$ denotes that the variables $Z_H/Z_R/Z_T$ can influence the data generation process of $H/R/T$.

The causal graph in Figure 1(b) shows that $Z_H/Z_R/Z_T$ simultaneously affects $H/R/T$ and $Y$, so $Z_H/Z_R/Z_T$ are confounders. The confounder $Z_H$ leads to two paths from $Z_H$ to $Y$: $Z_H \rightarrow Y$ and $Z_H \rightarrow H \rightarrow Y$. The first path combines edges $\{H, R, T\} \rightarrow Y$ to model $Y$, as expected. For example, if $Z_H$ denotes the popularity of persons, then persons with high $Z_H$ are more likely to have relation *is_friend_of* with others. Therefore, $Z_H$ is also a cause of $Y$. The second path means that $Z_H$ can affect the data generation of $H$. For example, if $Z_H$ denotes the popularity of head entities, then $Z_H$ will influence the likelihood of head entities being collected, making the collected data biased toward popular head entities (Mohamed et al., 2020). This causes bias amplification, which should be avoided because a KGC model should faithfully estimate the likelihood of triplets and not be affected by the way of data collection. The confounders $Z_R$ and $Z_T$ are similar to $Z_H$. Thus, the bad effect caused by the paths $Z_H \rightarrow H$, $Z_R \rightarrow R$ and $Z_T \rightarrow T$ should be blocked.

**Deconfounded KGC Models**   To eliminate the bad effect of confounders $\{Z_H, Z_R, Z_T\}$, we should rule out the paths $Z_H \rightarrow H$, $Z_R \rightarrow R$ and $Z_T \rightarrow T$ from the causal graph in Figure 1(b), that is exactly $do(H, R, T)$ operator, which results the causal graph in Figure 1(c). Thus, the causal effect of $\{H, R, T\}$ on $Y$ can be measured by probability function $P(Y|do(H, R, T))$ in Figure 1(b), i.e., $P(Y|H, R, T)$ in Figure 1(c). To estimate $P(Y|do(H, R, T))$, one method is to conduct randomized experiments. During the data collection process, we randomly select head entities, relations and tail entities, and then judge whether the corresponding triplets are true. This can make $\{H, R, T\}$ unaffected by confounders $\{Z_H, Z_R, Z_T\}$. However, randomized experiments are difficult to conduct. On the one hand, only the data collector can decide how the data is collected. On the other hand, since the triplets are obtained indirectly from unstructured text by algorithms, even the data collectors may not be able to manipulate the way of data collection. Therefore, it is crucial to estimate $P(Y|do(H, R, T))$ from only the observation data. Our method is to first convert $P(Y|do(H, R, T))$ into a statistical estimate. Then the statistical estimate can be obtained from the observation data.

The backdoor adjustment enables us to achieve it (Pearl, 2009b). The variables $\{Z_H, Z_R, Z_T\}$ satisfy the backdoor criterion because they block all the backdoor paths from variables $\{H, R, T\}$ to variable $Y$. Then $P(Y = y|do(H = h, R = r, T = t))$ can be obtained with backdoor adjustment

as follows:

$$P(Y = y|do(H = h, R = r, T = t)) = \sum_{z_h \in Z_H, z_r \in Z_R, z_t \in Z_T} P(y|h, r, t, z_h, z_r, z_t)P(z_h, z_r, z_t)$$

Thus, we can first estimate $P(y|h, r, t, z_h, z_r, z_t)$ from the observation data, which is feasible. Then we compute $P(y|do(h, r, t))$ by backdoor adjustment formula. $P(y|h, r, t, z_h, z_r, z_t)$ is relevant to the triplet $(h, r, t)$ and the confounders $(z_h, z_r, z_t)$. Thus, we need to define the confounders $(z_h, z_r, z_t)$. The confounders are reflected in the observation data, so they are functions of a 3rd-order binary tensor $\boldsymbol{X}$. We define two types of confounders as follows:

1. Artificially designed confounders:

$$z_h = \ln(\boldsymbol{1}^T \text{vec}(\boldsymbol{X}_{h,:,:})) = \ln(|\{(h_0, r_0, t_0) \in \mathcal{D}|h_0 = h\}|) = \ln(d(h))$$
$$z_r = \ln(\boldsymbol{1}^T \text{vec}(\boldsymbol{X}_{:,r,:})) = \ln(|\{(h_0, r_0, t_0) \in \mathcal{D}|r_0 = r\}|) = \ln(d(r))$$
$$z_t = \ln(\boldsymbol{1}^T \text{vec}(\boldsymbol{X}_{:,:,t})) = \ln(|\{(h_0, r_0, t_0) \in \mathcal{D}|t_0 = t\}|) = \ln(d(t))$$

where $\boldsymbol{1}$ is a vector of appropriate size whose elements are all 1 and $\text{vec}(\cdot)$ is an operation that expands a tensor into a vector. We define $d(h)/d(r)/d(t)$ as the popularity of a head entity $h$/a relation $r$/a tail entity $t$. The logarithmic function is to prevent the training unstable for the case where $d(h)/d(r)/d(t)$ is too large. The computational complexity of $z_h/z_r/z_t$ is $\mathcal{O}(1)$ because we can compute the value of $z_h/z_r/z_t$ in advance.

2. Learnable confounders: $z_h = q_H(\boldsymbol{X}_{h,:,:})$, $z_r = q_R(\boldsymbol{X}_{:,r,:})$, $z_t = q_T(\boldsymbol{X}_{:,:,t})$

where $q_H(\cdot)/q_R(\cdot)/q_T(\cdot)$ can be a $L$-layer neural network. We implement $q_H(\cdot)$ as a one-layer neural network for efficiency, i.e. $q_H(\boldsymbol{X}_{h,:,:}) = \boldsymbol{W}_H^T \text{vec}(\boldsymbol{X}_{h,:,:}) + \boldsymbol{b}_H$, where $\boldsymbol{W}_H$ is the weight matrix and $\boldsymbol{b}_H$ is the bias vector. $q_R(\cdot)$ and $q_T(\cdot)$ are similar to $q_H(\cdot)$. Then the computational complexity of $z_h/z_r/z_t$ is $\mathcal{O}(d(h))/\mathcal{O}(d(r))/\mathcal{O}(d(t))$.

Let $f(h, r, t)$ be a scoring function of a traditional KGC model. We define $P(y|h, r, t, z_h, z_r, z_t)$ as

$$P(y|h, r, t, z_h, z_r, z_t) \propto g(h, r, t, z_h, z_r, z_t) = g(f(h, r, t), z_h, z_r, z_t)$$
$$= f(h, r, t) + \alpha_h z_h + \alpha_r z_r + \alpha_t z_t$$

where $\alpha_h$, $\alpha_r$ and $\alpha_t$ are hyper-parameters to control the weight of confounders. Since $P(y|h, r, t, z_h, z_r, z_t)$ is used to rank tail entities, we do not need to normalize $P(y|h, r, t, z_h, z_r, z_t)$ to make it a rigorous probability function. In principle, $g(\cdot)$ can be any function, we implement $g(\cdot)$ as the addition of $f(h, r, t)$ and $\alpha_h z_h + \alpha_r z_r + \alpha_t z_t$ for simplicity. $g(h, r, t, z_h, z_r, z_t)$ can be seen as the new scoring function from the old scoring function $f(h, r, t)$. If $\alpha_h = \alpha_r = \alpha_t = 0$, $g(h, r, t, z_h, z_r, z_t)$ reduces to the traditional scoring function $f(h, r, t)$.

Now we can train a model to get $g(h, r, t, z_h, z_r, z_t)$. We use the multi-class loss function as in (Lacroix et al., 2018). For a training triplet $(h, r, t)$, our loss function is

$$\ell(g(h, r, t, z_h, z_r, z_t)) = -g(h, r, t, z_h, z_r, z_t) + \log(\sum_{t'=1}^{|\mathcal{E}|} \exp(g(h, r, t', z_h, z_r, z_{t'})))$$

where $\{z_h, z_r, z_{t'}\}$ is the value of confounding variables corresponding to the triplet $(h, r, t')$.

After getting $g(h, r, t, z_h, z_r, z_t)$, we can compute $P(Y = y|do(H = h, R = r, T = t))$ by backdoor adjustment:

$$y = P(Y = y|do(H = h, R = r, T = t))$$
$$= \sum_{z_h \in Z_H, z_r \in Z_R, z_t \in Z_T} P(y|h, r, t, z_h, z_r, z_t)P(z_h, z_r, z_t)$$
$$\propto \sum_{z_h \in Z_H, z_r \in Z_R, z_t \in Z_T} g(h, r, t, z_h, z_r, z_t)P(z_h, z_r, z_t)$$
$$= \sum_{z_h \in Z_H, z_r \in Z_R, z_t \in Z_T} (f(h, r, t) + \alpha_h z_h + \alpha_r z_r + \alpha_t z_t)P(z_h, z_r, z_t)$$
$$= f(h, r, t) + \sum_{z_h \in Z_H, z_r \in Z_R, z_t \in Z_T} (\alpha_h z_h + \alpha_r z_r + \alpha_t z_t)P(z_h, z_r, z_t)$$

Since $\sum_{z_h \in Z_H, z_r \in Z_R, z_t \in Z_T} (\alpha_h z_h + \alpha_r z_r + \alpha_t z_t) P(z_h, z_r, z_t)$ is equal for all triplets $(h, r, t)$, we can use $f(h, r, t)$ to surrogate $P(Y = y | do(H = h, R = r, T = t))$.

Our final result is easy, we only need to add an extra term $\alpha_h z_h + \alpha_r z_r + \alpha_t z_t$ to the traditional KGC model $f(h, r, t)$ in the training process and then get the deconfounded/causal KGC model $f(h, r, t)$. Our method is model-agnostic, which can be appilied to any traditional KGC model.

### 3.2 EVALUATION OF CAUSAL KGC MODELS

Traditional KGC models are trained on a set of true triplets and evaluated on holdout test triplets. The ranking metrics, MRR and H@N (Bordes et al., 2013), are often used to evaluate the KGC models. The definitions of MRR and H@N are as follows:

MRR $= \sum_{(h,r,t) \in \mathcal{D}} \frac{1}{|\mathcal{D}|} \frac{1}{\text{rank}(h,r,t)}$, where $\text{rank}(h, r, t)$ is the rank of tail entity $t$ in the predicted list for the query $(h, r, ?)$. Higher MRR indicates better performance.

H@N $= \sum_{(h,r,t) \in \mathcal{D}} \frac{1}{|\mathcal{D}|} \mathbb{I}(\text{rank}(h, r, t) \leq N)$, where $\mathbb{I}(\cdot)$ is the indicator function. H@N is the ratio of the ranks that no more than $N$. Higher H@N indicates better performance.

However, causal KGC models present unique challenges for evaluation. Which test set should we use to evaluate causal KGC models? If we evaluate the models on the observation test set, the result gives a biased evaluation: it favours popular entities and relations. One solution is to evaluate on a randomized test set. However, a randomized test set is often difficult to obtain. Another solution is to evaluate the models on the observation test set with new evaluation metrics. Mohamed et al. (2020) propose a new evaluation metric based on the popularity of entities and relations. Similar to (Mohamed et al., 2020), we define a weighted evaluation metric W-Metric$(\beta_h, \beta_r, \beta_t)$ as

$$\text{W-Metric}(\beta_h, \beta_r, \beta_t) = \sum_{(h,r,t) \in \mathcal{D}} w(h, r, t) u(h, r, t)$$

where $w(h, r, t) = \frac{d(h)^{\beta_h} d(r)^{\beta_r} d(t)^{\beta_t}}{\sum_{(h,r,t) \in \mathcal{D}} d(h)^{\beta_h} d(r)^{\beta_r} d(t)^{\beta_t}}$, $\sum_{(h,r,t) \in \mathcal{D}} w(h, r, t) = 1$, $d(h)/d(r)/d(t)$ is the popularity of $h/r/t$ and $u(h, r, t) = \frac{1}{\text{rank}(h,r,t)}$ or $u(h, r, t) = \mathbb{I}(\text{rank}(h, r, t) \leq N)$.

We denote W-Metric$(\beta_h, \beta_r, \beta_t)$ as W-MRR$(\beta_h, \beta_r, \beta_t)$ if $u(h, r, t) = \frac{1}{\text{rank}(h,r,t)}$ and denote W-Metric$(\beta_h, \beta_r, \beta_t)$ as W-H@N$(\beta_h, \beta_r, \beta_t)$ if $u(h, r, t) = \mathbb{I}(\text{rank}(h, r, t) \leq N)$.

Let $\beta_h = \beta_r = \beta_t = 0$, then W-Metric$(\beta_h, \beta_r, \beta_t) = $ W-Metric$(0, 0, 0) = $MRR or H@N. Thus, W-Metric$(\beta_h, \beta_r, \beta_t)$ can be seen as a generalization of MRR or H@N. Let $\beta_h = -1, \beta_r = 0, \beta_t = 0$, then

$$\text{W-Metric}(\beta_h, \beta_r, \beta_t) = \text{W-Metric}(-1, 0, 0) = \sum_{(h,r,t) \in \mathcal{D}} \frac{1/d(h)}{\sum_{(h,r,t) \in \mathcal{D}} 1/d(h)} u(h, r, t)$$

$$= \sum_{h \in \mathcal{E}, d(h) > 0} \frac{1}{|\{h \in \mathcal{E} | d(h) > 0\}|} \sum_{(h_0, r_0, t_0) \in \mathcal{D}, h_0 = h} \frac{u(h, r_0, t_0)}{d(h)}$$

$$= \sum_{h \in \mathcal{E}, d(h) > 0} \frac{v(h)}{|\{h \in \mathcal{E} | d(h) > 0\}|}$$

where $v(h) = \sum_{(h_0, r_0, t_0) \in \mathcal{D}, h_0 = h} \frac{u(h, r_0, t_0)}{d(h)}$. For each head entity $h$, W-Metric$(-1, 0, 0)$ first computes the mean of $u(h, r_0, t_0)$, i.e., $v(h)$. While popular head entities receive more $u(h, r_0, t_0)$, W-Metric$(-1, 0, 0)$ treats all head entities equally, regardless of the popularity $d(h)$. This can eliminate the influence of $d(h)$. W-Metric$(-1, 0, 0)$ then computes the mean of $v(h)$ of all head entities. Thus, W-Metric$(-1, 0, 0)$ is to evaluate the average per-head-entity $v(h)$.

Similarly, W-Metric$(0, -1, 0)$/W-Metric$(0, 0, -1)$ is to evaluate the average per-relation/per-tail-entity $v(r)/v(t)$. Combining these three metrics, we evaluate causal KGC models with W-Metric$(-1, -1, -1)$, which simultaneously takes into account the popularity of head entities, relations and tail entities. For every triplet $(h, r, t)$, W-Metric$(-1, -1, -1)$ use $\frac{1/(d(h)d(r)d(t))}{\sum_{(h,r,t) \in \mathcal{D}} 1/(d(h)d(r)d(t))}$ to weight $u(h, r, t)$.

### 3.3 A Group Theory View for KGC

The main feature of causal is invariance or symmetry (Arjovsky et al., 2020). Group theory is a mathematical language to describe symmetry. Thus, we show a group theory view for KGC, which is equivalent to the view of causal but further reveals the causal relationships.

Let $\mathcal{X} = \mathcal{E} \times \mathcal{R} \times \mathcal{E}$, a KGC model is to find a scoring function $f(x)$ that holds for all triplets $x \in \mathcal{X}$, i.e., the form of $f(x)$ should be invariant for all $x \in \mathcal{X}$. We next show the invariance of the form of $f(x)$ can be associated with the notations of groups. We define a group action $\gamma$ of $S_{\mathcal{X}}$ on $\mathcal{X}$ as $\gamma(a, x) = a(x)$, where $a \in S_{\mathcal{X}}, x \in \mathcal{X}$ and $S_{\mathcal{X}}$ is the the symmetry group of $\mathcal{X}$. Then the orbit of $x \in \mathcal{X}$ is $S_{\mathcal{X}} \cdot x = \{\gamma(a, x) | a \in S_{\mathcal{X}}\}$, which is exactly equal to $\mathcal{X}$. Thus, the form of $f(x)$ should be invariant for all $x \in S_{\mathcal{X}} \cdot x$. Now we have established the correspondence between $f(x)$ and a group $S_{\mathcal{X}}$. We say a scoring function $f(x)$ satisfies $G$ invariance if the form of $f(x)$ is invariant for all $x \in G \cdot x$, where $G$ is a subgroup of $S_{\mathcal{X}}$. Thus, traditional KGC models are to learn a $S_{\mathcal{X}}$ invariance scoring function. The causal graph in Figure 1(a) can correspond to a group $S_{\mathcal{X}}$.

If we have all the data, we can obviously learn the correct $S_{\mathcal{X}}$ invariance scoring function $f(x)$. However, we only have some of the data, which may not match the overall data due to the data bias. Thus, we should not treat all $x \in S_{\mathcal{X}} \cdot x$ equally, i.e., treat only some $x \in S_{\mathcal{X}} \cdot x$ equally. Therefore, the scoring function learned from data should satisfy $G$ (a subgroup of $S_{\mathcal{X}}$) invariance. For example, if $G = S_{\{h\} \times \mathcal{R} \times \mathcal{E}}$, then learning $G$ invariance scoring function can correspond to learning $P(y|h, r, t, z_h)$ in causal KGC models. After we learn the $G$ invariance scoring function, we can recover the $S_{\mathcal{X}}$ invariance scoring function by using the quotient group $S_{\mathcal{X}}/G$ to act on $P(y|h, r, t, z_h)$. This can correspond to computing $P(y|do(H = h, R = r, T = t)) = \sum_{z_h \in Z_H} P(y|h, r, t, z_h)P(z_h)$ in causal KGC models.

In summary, we want to learn a $S_{\mathcal{X}}$ invariance scoring function $f(x)$. However, the biased data only allows us to learn a $G$ invariance scoring function. In order to recover $S_{\mathcal{X}}$ invariance, we can use the quotient group $S_{\mathcal{X}}/G$ to act on the $G$ invariance scoring function to get the $S_{\mathcal{X}}$ invariance scoring function $f(x)$.

The advantages of the view of group theory to the view of causal are in three folds. First, some relationships between variables are hard to describe by causal graphs. For example, for the ideal gas law $PV = nRT$, it is hard to say what causes what (Arjovsky et al., 2020). In group theory, we use the invariance to represent the relationships between variables, which is easy to describe. Second, there is no metric to measure the relationships between causal graphs. In group theory, we use the notations of group theory to measure the relationships between groups, such as the order of groups, normal subgroups etc., which help us understand the invariance. Third, backdoor adjustment formula is only suitable for probability function. In group theory, we can use the group to act on any function. Thus, the view of group theory generalizes backdoor adjustment formula to any function.

The potential applications of the view of group theory are in three folds. First, the view of group theory enables KGC models to solve out-of-distribution problem. In this paper, we learn a $G$ invariance scoring function from the biased data and apply a group action to the $G$ invariance scoring function to make it able to evaluate on unbiased data. Thus, the task of this paper can be seen as an out-of-distribution task. We can apply different group actions to the $G$ invariance scoring function to make it adapt to different test sets. Second, learning $G$ invariance scoring function directly may be difficult, the direct product decomposition and subgroup series allow us to simplify the group $G$. Third, the $G$ invariance is pre-specified in this paper. It is worth exploring how to learn the invariance from data automatically? Given a group action on $\mathcal{X}$, the orbits form a partition of $\mathcal{X}$. Thus, we can transform learning the invariance into learning the orbits of $\mathcal{X}$.

## 4 Related Work

Data bias refers to data that does not reflect the true distribution. Although the data bias problem has been extensively studied in many machine learning fields, such as imbalanced classification problem (Krawczyk, 2016) and data bias in recommendation systems (Chen et al., 2020), there are few works considering data bias in KGs. Mohamed et al. (2020) show that benchmark datasets suffer from the popularity bias and existing KGC models are biased towards popular entities and relations. Bonner et al. (2022) show the existence of popularity bias of entities in biomedical KGs.

Table 1: Knowledge graph completion results on FB15k-237, WN18RR and YGAO3-10 test sets with evaluation metrics **W-Metric(-1, -1, -1)**.

| | FB15k-237 | | | WN18RR | | | YAGO3-10 | | |
|---|---|---|---|---|---|---|---|---|---|
| | MRR | H@1 | H@10 | MRR | H@1 | H@10 | MRR | H@1 | H@10 |
| TransE | 0.279 | 0.168 | 0.509 | 0.224 | 0.061 | 0.568 | 0.303 | 0.205 | 0.480 |
| IPS-TransE | **0.307** | **0.210** | 0.512 | **0.279** | **0.078** | **0.566** | 0.299 | 0.196 | **0.498** |
| Causal-TransE-1 | 0.283 | 0.166 | **0.514** | 0.227 | 0.062 | 0.565 | **0.307** | **0.207** | 0.488 |
| Causal-TransE-2 | 0.285 | 0.175 | 0.507 | 0.225 | 0.058 | 0.559 | 0.303 | 0.203 | 0.484 |
| RotatE | 0.274 | 0.177 | 0.477 | 0.514 | 0.479 | 0.581 | 0.330 | 0.237 | 0.496 |
| IPS-RotatE | 0.281 | **0.191** | 0.470 | 0.502 | 0.475 | 0.553 | 0.260 | 0.202 | 0.385 |
| Causal-RotatE-1 | **0.288** | 0.186 | **0.493** | **0.525** | **0.487** | **0.603** | **0.364** | **0.282** | **0.529** |
| Causal-RotatE-2 | 0.280 | 0.189 | 0.465 | 0.521 | 0.485 | 0.600 | 0.341 | 0.264 | 0.496 |
| DistMult | 0.302 | 0.195 | 0.514 | 0.521 | 0.478 | 0.607 | 0.362 | 0.284 | 0.535 |
| IPS-DistMult | 0.304 | 0.208 | 0.501 | 0.522 | **0.480** | 0.611 | 0.328 | 0.270 | 0.445 |
| Causal-DistMult-1 | 0.312 | 0.206 | 0.533 | **0.523** | 0.478 | **0.617** | **0.382** | **0.295** | **0.550** |
| Causal-DistMult-2 | **0.325** | **0.217** | **0.553** | 0.521 | 0.477 | 0.608 | 0.362 | 0.284 | 0.537 |
| ComplEx | 0.307 | 0.204 | 0.517 | **0.535** | **0.493** | 0.619 | 0.370 | 0.278 | 0.573 |
| IPS-ComplEx | 0.307 | 0.205 | 0.511 | 0.532 | 0.488 | **0.623** | 0.335 | 0.263 | 0.473 |
| Causal-ComplEx-1 | **0.330** | **0.223** | **0.555** | **0.535** | **0.493** | 0.617 | **0.386** | **0.292** | **0.586** |
| Causal-ComplEx-2 | 0.323 | 0.220 | 0.541 | 0.532 | 0.491 | 0.611 | 0.375 | 0.286 | 0.567 |

We utilize causal graphs to make KGC models more explainable. Existing works on explainable KGC models mainly focus on combining embedding with symbolic reasoning (Zhang et al., 2021a). Guo et al. (2016) utilize the logical rules to improve the performance of embedding-based methods. Markov logic network (Richardson & Domingos, 2006) designs a probabilistic framework to represent the logical rules as features. Neural LP (Yang et al., 2017) derives the logical rules from data and leverages neural networks to deal with the uncertainty and ambiguity of data.

Causal inference techniques have been used to alleviate the data bias issue (Gao et al., 2022), including SCMs and potential outcome framework. Under SCMs, data bias can be seen as confounders, then backdoor adjustment (Zhang et al., 2021b) and frontdoor adjustment (Xu et al., 2021) are utilized to eliminate the confounding effect. Based on the potential outcome framework, Schnabel et al. (2016) propose the inverse propensity score (IPS) method, which aims to reweight the samples by the chances that they receive the treatments.

## 5 EXPERIMENTS

We first introduce the experimental settings. Then, we show the results of our causal KGC models and compare with other models. Finally, we conduct ablation studies. Please see Appendix A.1 for more experimental details.

### 5.1 EXPERIMENTAL SETTINGS

**Datasets** We evaluate the models on three popular KGC datasets, FB15k-237 (Toutanova et al., 2015), WN18RR (Dettmers et al., 2018) and YAGO3-10 (Dettmers et al., 2018).

**Models** We use original TransE (Bordes et al., 2013), RotatE (Sun et al., 2018), DistMult (Yang et al., 2014) and ComplEx (Toutanova et al., 2015) as baselines. We denote TransE with IPS (Schnabel et al., 2016) method as IPS-TransE. We denote TransE with our causal method as Causal-TransE. We denote Causal-TransE with artificially designed confounders as Causal-TransE-1 and TransE with learnable confounders as Causal-TransE-2. The notations of RotatE, DistMult and ComplEx are similar to TransE.

Table 2: The results on FB15k-237 datasets with different hyper-parameters.

| | Causal-ComplEx-1 | | | Causal-ComplEx-2 | | |
|---|---|---|---|---|---|---|
| | MRR | H@1 | H@10 | MRR | H@1 | H@10 |
| $\alpha_h = 0, \alpha_r = 0, \alpha_t = 0$ | 0.307 | 0.204 | 0.517 | 0.307 | 0.204 | 0.517 |
| $\alpha_h \neq 0, \alpha_r = 0, \alpha_t = 0$ | 0.308 | 0.210 | 0.519 | 0.307 | 0.204 | 0.517 |
| $\alpha_h = 0, \alpha_r \neq 0, \alpha_t = 0$ | 0.308 | 0.211 | 0.528 | 0.308 | 0.211 | 0.528 |
| $\alpha_h = 0, \alpha_r = 0, \alpha_t \neq 0$ | **0.333** | **0.227** | 0.553 | 0.327 | 0.218 | **0.558** |
| $\alpha_h \neq 0, \alpha_r \neq 0, \alpha_t \neq 0$ | 0.330 | 0.223 | **0.555** | 0.323 | 0.220 | 0.541 |
| $\alpha_h = \alpha_t$ | 0.327 | 0.222 | 0.551 | 0.326 | **0.222** | **0.558** |
| $\alpha_h = \alpha_r = \alpha_t$ | 0.322 | 0.216 | 0.553 | **0.328** | 0.218 | 0.557 |

**Evaluation Metrics** We use W-MRR(-1,-1,-1) and W-H@N(-1,-1,-1) as evaluation metrics and choose the hyper-parameters with the best W-MRR(-1,-1,-1) on the validation set.

## 5.2 RESULTS

The results show that our causal KGC models achieve improvement on different datasets, different models and different evaluation metrics (MRR, H@1 and H@10). This demonstrates the effectiveness of our causal KGC models. Models with artificially designed confounders overall is better than models with learnable confounders. The reason is that models with learnable confounders are more difficult to optimize and are more likely to overfit.

Our models are better than the IPS method. IPS method do not achieve consistent performance due to the difficulty of estimating propensity score. The improvement of our causal KGC models is significant on FB15k-237 dataset and YAGO3-10 dataset, and is little on WN18RR dataset. The reason is that the degree of data bias on WN18RR dataset is smaller, as shown in Table 4.

## 5.3 ABLATION STUDIES

We conduct ablation studies to analyze which of the confounders $\{Z_H, Z_R, Z_T\}$ influences the models most. We use the models with $\alpha_h = \alpha_r = \alpha_t = 0$ as baselines. We train models with only one of the hyper-parameters $\{\alpha_h, \alpha_r, \alpha_t\}$. We also train models that use fewer hyper-parameters by setting $\alpha_h = \alpha_t$ and $\alpha_h = \alpha_r = \alpha_t$. Since $Z_h$ and $Z_t$ are similar, one is a confounder of head entities and the other one is a confounder of tail entities, we set $\alpha_h = \alpha_t$. All experiments are trained on FB15k-237 dataset with ComplEx model. See Table 2 for the results.

The results show that $Z_T$ influences models most, $Z_H$ and $Z_R$ influence models little. The reason is that KGC is formulated as predicting the tail entities. The performance of the models have no obvious attenuation if we set $\alpha_h = \alpha_t$ or $\alpha_h = \alpha_r = \alpha_t$. Thus, we can reduce the computation by using fewer hyper-parameters.

## 6 CONCLUSION

Traditional KGC models only consider the correlation in the data and ignore the causation, which leads to the data bias issues. In this paper, we utilize causal inference to alleviate the data bias issues. Some research directions on how to apply causal inference to KGC deserve further thought.

First, we suppose that the confounders $\{Z_H, Z_R, Z_T\}$ in the causal graph of Figure 1(b) affect $\{H, R, T\}$ individually. Confounders that affect at least two of $\{H, R, T\}$ at the same time or other types of confounders are worth considering. Second, for the learnable confounders, deeper neural networks are worth exploring. Third, we use backdoor adjustment to get our causal KGC models, how to use frontdoor adjustment in KGC models is also worth exploring. Fourth, counterfactual reasoning is another technique of causal inference, which can be used to augment KG data.

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

## A  APPENDIX

### A.1  EXPERIMENTAL DETAILS

**Datasets** We evaluate our models on three popular knowledge graph completion datasets, WN18RR Dettmers et al. (2018), FB15k-237 Toutanova et al. (2015) and YAGO3-10 Dettmers et al. (2018). WN18RR is a subset of WN18, with inverse relations removed. WN18 is extracted from WordNet, a database containing lexical relations between words. FB15k-237 is a subset of FB15k, with inverse relations removed. FB15k is extracted from Freebase, a large database of real world facts. YAGO3-10 is a subset of YAGO3 that only contains entities with at least 10 relations. The statistics of the datasets are shown in Table 3.

Table 3: The statistics of the datasets.

| Dataset | #entity | #relation | #training | #validation | #test |
|---|---|---|---|---|---|
| WN18RR | 40,943 | 11 | 86,835 | 3,034 | 3,134 |
| FB15k-237 | 14,541 | 237 | 272,115 | 17,535 | 20,466 |
| YGAO3-10 | 123,188 | 37 | 1,079,040 | 5,000 | 5,000 |

**Data Bias in Datasets** We use the Gini coefficients of $d(h)/d(r)/d(t)$ to measure the data bias of $h/r/t$ in the dataset $\mathcal{D}$, denote as $g_h(\mathcal{D})/g_r(\mathcal{D})/g_t(\mathcal{D})$. Denote the training set as $\mathcal{D}_1$, validation set as $\mathcal{D}_2$, and test set as $\mathcal{D}_3$. The Gini coefficients of WN18RR, FB15k-237 and YAGO3-10 datasets are shown in Table 4. Gini coefficients are not less than 0 and not larger than 1. Larger Gini coefficients mean larger data bias. The results show that the degree of data bias on WN18RR dataset is less than that on FB15k-237 dataset or YAGO3-10 dataset.

**Hyper-parameters** We add the N3 regularization (Lacroix et al., 2018) to the loss function and let the regularization coefficient be $\lambda$. We set the batch size to 1024, epoch to 50 and embedding dimension to 2048 for all models. We use Adam Kingma & Ba (2014) with exponential decay as the optimizer. We search the learning rate in $\{0.001, 0.003, 0.005, 0.01\}$, decay rate in $\{0.9, 0.93, 0.95, 1.0\}$, $\beta_1, \beta_2, \beta_3$ in $\{0.1, 0.3, 0.5, 0.7, 0.9\}$, $\lambda$ in $\{0.0, 0.001, 0.003, 0.01, 0.03\}$. We first set $\beta_1 = \beta_2 = \beta_3 = \lambda = 0$ and search the learning rate and decay rate. For WN18RR dataset, we set learning rate to 0.01 and decay rate to 0.9 for all models. For FB15k-237 dataset, we set learning rate to 0.005 and decay rate to 0.93 for all models. For YAGO3-10 dataset, we set learning rate to 0.003 and decay rate to 0.9 for all models. We search at most 50 hyper-parameters combinations.

Table 4: The Gini coefficients of WN18RR, FB15k-237 and YAGO3-10 datasets. Larger Gini coefficients mean larger data bias.

| Dataset | $g_h(\mathcal{D}_1)$ | $g_r(\mathcal{D}_1)$ | $g_t(\mathcal{D}_1)$ | $g_h(\mathcal{D}_2)$ | $g_r(\mathcal{D}_2)$ | $g_t(\mathcal{D}_2)$ | $g_h(\mathcal{D}_3)$ | $g_r(\mathcal{D}_3)$ | $g_t(\mathcal{D}_3)$ |
|---------|------|------|------|------|------|------|------|------|------|
| WN18RR | 0.453 | 0.667 | 0.453 | 0.136 | 0.664 | 0.136 | 0.139 | 0.664 | 0.139 |
| FB15k-237 | 0.560 | 0.679 | 0.560 | 0.511 | 0.718 | 0.511 | 0.519 | 0.718 | 0.519 |
| YGAO3-10 | 0.573 | 0.832 | 0.574 | 0.191 | 0.816 | 0.191 | 0.191 | 0.811 | 0.191 |

**Learnable Confounders** We design the learnable confounders $z_h = q_H(\boldsymbol{X}_{h,:,:}) = \boldsymbol{W}_H^T \text{vec}(\boldsymbol{X}_{h,:,:}) + \boldsymbol{b}_H$, $z_r = q_R(\boldsymbol{X}_{:,r,:}) = \boldsymbol{W}_R^T \text{vec}(\boldsymbol{X}_{:,r,:}) + \boldsymbol{b}_R$ and $z_t = q_T(\boldsymbol{X}_{:,:,t}) = \boldsymbol{W}_T^T \text{vec}(\boldsymbol{X}_{:,:,t}) + \boldsymbol{b}_T$. The number of parameters of $\boldsymbol{W}_H/\boldsymbol{W}_T$ is the product of the number of entities and the number of relations. The number of parameters of $\boldsymbol{W}_R$ is the square of the number of entities, which is too large for large datasets. Thus, we use a weight sharing method to reduce the number of parameters to a reasonable size. The number of parameters of $\boldsymbol{b}_H/\boldsymbol{b}_R/\boldsymbol{b}_T$ is 1.

**Inverse Propensity Scoring** We use the inverse propensity scoring (IPS) method as a baseline method. Thus, we show a more detailed statement about IPS. IPS method reweights the weight of the loss function for a batch of data $(h_i, r_i, t_i)$, where $1 \le i \le B$ and $B$ is the batch size. The loss function is defined as $L = \sum_{i=1}^{B} w_i \ell(g(h_i, r_i, t_i, z_{h_i}, z_{r_i}, z_{t_i}))$, where

$$w_i = \frac{d(h_i)^{-1} d(r_i)^{-1} d(t_i)^{-1}}{\sum_{i=1}^{B} d(h_i)^{-1} d(r_i)^{-1} d(t_i)^{-1}}$$

is the weight (i.e., the inverse propensity score) and

$$\ell(g(h_i, r_i, t_i, z_{h_i}, z_{r_i}, z_{t_i})) = -g(h_i, r_i, t_i, z_{h_i}, z_{r_i}, z_{t_i}) + \log(\sum_{t_i'=1}^{|\mathcal{E}|} \exp(g(h_i, r_i, t_i', z_{h_i}, z_{r_i}, z_{t_i'})))$$

where $\{z_{h_i}, z_{r_i}, z_{t_i'}\}$ is the value of confounding variables corresponding to the triplet $(h_i, r_i, t_i')$.

Our causal KGC model adds an extra term $\alpha_h z_h + \alpha_r z_r + \alpha_t z_t$ to the traditional KGC model $f(h, r, t)$. Thus, IPS methods focus on the loss function, our causal KGC model focuses on the scoring function.

**Sensitivity Analysis** We analysis the sensitivity of our models with respect to the hyper-parameters $\alpha_h, \alpha_r$ and $\alpha_t$. We run experiments by setting $\alpha_h$ to $\{0.0, 0.2, 0.4, 0.6, 0.8, 1.0\}$ and $\alpha_r = \alpha_t = 0.0$ to analysis the sensitivity of models with respect to $\alpha_h$. The hyper-parameters $\alpha_r$ and $\alpha_t$ are similar to the hyper-parameter $\alpha_h$. All experiments are trained on FB15k-237 dataset with ComplEx model. See Table 5 for the results. The results show that our models are not sensitive to the hyper-parameters $\alpha_h, \alpha_r$ and $\alpha_t$.

Table 5: The results on FB15k-237 datasets with different hyper-parameters.

| | Causal-ComplEx-1 | | | Causal-ComplEx-2 | | |
|---|---|---|---|---|---|---|
| | MRR | H@1 | H@10 | MRR | H@1 | H@10 |
| $\alpha_h = 0.0, \alpha_r = 0.0, \alpha_t = 0.0$ | 0.307 | 0.204 | 0.517 | 0.307 | 0.204 | 0.517 |
| $\alpha_h = 0.2, \alpha_r = 0.0, \alpha_t = 0.0$ | 0.318 | 0.220 | 0.532 | 0.303 | 0.201 | 0.517 |
| $\alpha_h = 0.4, \alpha_r = 0.0, \alpha_t = 0.0$ | 0.306 | 0.210 | 0.509 | 0.310 | 0.209 | 0.532 |
| $\alpha_h = 0.6, \alpha_r = 0.0, \alpha_t = 0.0$ | 0.311 | 0.207 | 0.533 | 0.306 | 0.200 | 0.529 |
| $\alpha_h = 0.8, \alpha_r = 0.0, \alpha_t = 0.0$ | 0.310 | 0.211 | 0.512 | 0.309 | 0.204 | 0.520 |
| $\alpha_h = 1.0, \alpha_r = 0.0, \alpha_t = 0.0$ | 0.298 | 0.189 | 0.529 | 0.305 | 0.200 | 0.534 |
| $\alpha_h = 0.0, \alpha_r = 0.0, \alpha_t = 0.0$ | 0.307 | 0.204 | 0.517 | 0.307 | 0.204 | 0.517 |
| $\alpha_h = 0.0, \alpha_r = 0.2, \alpha_t = 0.0$ | 0.302 | 0.195 | 0.536 | 0.308 | 0.204 | 0.527 |
| $\alpha_h = 0.0, \alpha_r = 0.4, \alpha_t = 0.0$ | 0.302 | 0.200 | 0.525 | 0.293 | 0.186 | 0.509 |
| $\alpha_h = 0.0, \alpha_r = 0.6, \alpha_t = 0.0$ | 0.309 | 0.206 | 0.524 | 0.308 | 0.209 | 0.522 |
| $\alpha_h = 0.0, \alpha_r = 0.8, \alpha_t = 0.0$ | 0.304 | 0.204 | 0.531 | 0.310 | 0.208 | 0.533 |
| $\alpha_h = 0.0, \alpha_r = 1.0, \alpha_t = 0.0$ | 0.301 | 0.189 | 0.529 | 0.307 | 0.204 | 0.516 |
| $\alpha_h = 0.0, \alpha_r = 0.0, \alpha_t = 0.0$ | 0.307 | 0.204 | 0.517 | 0.307 | 0.204 | 0.517 |
| $\alpha_h = 0.0, \alpha_r = 0.0, \alpha_t = 0.2$ | 0.323 | 0.208 | 0.556 | 0.321 | 0.220 | 0.532 |
| $\alpha_h = 0.0, \alpha_r = 0.0, \alpha_t = 0.4$ | 0.329 | 0.222 | 0.548 | 0.322 | 0.214 | 0.547 |
| $\alpha_h = 0.0, \alpha_r = 0.0, \alpha_t = 0.6$ | 0.332 | 0.229 | 0.551 | 0.326 | 0.220 | 0.552 |
| $\alpha_h = 0.0, \alpha_r = 0.0, \alpha_t = 0.8$ | 0.329 | 0.221 | 0.552 | 0.322 | 0.216 | 0.546 |
| $\alpha_h = 0.0, \alpha_r = 0.0, \alpha_t = 1.0$ | 0.323 | 0.215 | 0.557 | 0.323 | 0.213 | 0.539 |

