# OpenReview forum: "Causal Inference for Knowledge Graph Completion"
_ICLR.cc/2023/Conference — Submitted to ICLR 2023_

### Official Review · Reviewer_L8AG · 2022-10-24

**Confidence:** 3
**Clarity, Quality, Novelty And Reproducibility:** 1. The notation is abusive. For examp…
**Correctness:** 2
**Technical Novelty And Significance:** 2
**Empirical Novelty And Significance:** 2
**Recommendation:** 3

**Strength And Weaknesses:**

Strength: The authors evaluate the proposed causal KCG by relatively comprehensive empirical experiments.

Weaknesses:

1. The preliminaries and background introduction for the SCM models should be more comprehensive, as this is the fundamental point for the later KGC model development.

2. The logarithmic function definition for designed confounders is restrictive. Otherwise, the author should refer to the existing literature for such definition and evidence.

3. In the one-layer neural network, is it able to approximate the complex confounder? This seems to disobey the universal approximation theorem in DNN.

4. The score function $f(h,r,t)$ is not clear how affect the conditional probability $P(y|h,r,t,z_h,z_r,z_t)$. In addition, the linear decomposition might require the $g$ function is soothing. However, the authors seem not to address or mention this part in the paper.

5. The balancing of $\alpha_{h}, \alpha_{r}$ and $\alpha_{t}$ is crucial to balance the bias-variance tradeoff, the authors might be better to discuss this.

6. The authors define a novel metric for evaluation purposes. I was wondering if is there any existing metric for better comparison.

7. The causal-distmult-2 outperforms the competing methods in FB15k-237. But other ones, like causal TransE-1, RotatE-1, and ComplEX-1 are best. Could the authors give some analysis of this phenomenon?

**Summary Of The Paper:**

The knowledge graph completion is a key to studying the correlations in data. This paper proposes to study the causality of the KGC from a causal graph point of view. The method is evaluated by empirical experiments. However, in total, the novelty of the approach and the profound impact are both limited.



**Summary Of The Review:**

The paper studies the causality of KGC and uses numerical experiments to justify the proposed method. However, the development of the method is not clear and the paper is not well-presented. This limits the potential impact of the paper.

---

> ### Author Response · Authors · 2022-11-15
> **Rebuttal for Reviewer L8AG, Part One**
>
> Thank you for your careful and detailed comments. We have polished the paper for a clearer statement.
>
> Q1: The background introduction for SCM should be more comprehensive.
>
> A1: We have rewritten the background section to make it more comprehensive.
>
> Q2: The logarithmic function definition for designed confounders is restrictive.
>
> A2: The logarithmic function is $\textbf{not}$ restrictive. First, since the degree is positive, the logarithmic function is always valid. Second, since the logarithmic function is a monotonically increasing function (i.e., a one-to-one mapping), it can preserve the order. Third, the logarithmic function can prevent the training unstable for the case where the degree is too large. For example, the maximum of the degree of entities/relations is 482/34796 on WN18RR training set. Too large degree may lead to unstable training. Fourth, the hyper-parameters can also control the weight of confounders, which can make the value of confounders flexible.
>
> Q3: In the one-layer neural network, is it able to approximate the complex confounder?
>
> A3: First, it is a trade-off between expressiveness and computation. Deeper networks, more expressiveness and more computation. For one-layer neural network, the computational complexity of $z_{h}/z_{r}/z_{t}$ is $\mathcal{O}(d(h))/\mathcal{O}(d(r))/\mathcal{O}(d(t))$. For two-layer neural network, the computational complexity of $z_{h}/z_{r}/z_{t}$ is $\mathcal{O}(d(h)*D)/\mathcal{O}(d(r)*D)/\mathcal{O}(d(t)*D)$, where $D$ is the dimension of hidden features.
>
> Second, for two-layer networks, the number of parameters is about $D\*|\mathcal{E}|\*|\mathcal{R}|$, which is too large for large dataset. For example, for FB15k-237 dataset and $D=64$, the number of parameters is at least 4*10^8, which is about $D=64$ times of the number of parameters of one-layer networks.
>
> Third, deeper neural networks do not mean better performance. For example, the performance of ComplEx is better than most of the neural networks KGC models. Deeper neural networks are more difficult to train and more likely to overfit. In our experiments, the performance of models with artificially designed confounders overall is better than the performance of models with learnable confounders, though models with learnable confounders are more expressive. Thus, the complexity of models should match with the datasets. Deeper networks are worth exploring, but they may need some regularizations.
>
> Q4: The score function $f(h,r,t)$ is not clear how affect the conditional probability $P(y|h,r,t,z_{h},z_{r},z_{t})$. The smoothing of the function $g(\cdot)$ do not mention.
>
> A4: Since the evaluation metrics (MRR or H@N) are the ranking metrices, we do not need to normalize $P(y|h,r,t,z_{h},z_{r},z_{t})$ to make it a rigorous probability function. We can surrogate $P(y|hr,t,z_{h},z_{r},z_{t})$ by $g(h,r,t,z_{h},z_{r},z_{t})$, which keeps the ranking unchanged.
>
> Since $g(h,r,t,z_{h},z_{r},z_{t})$ is the addition of $f(h,r,t)$ and $\alpha_{h}z_{h}+\alpha_{r}z_{r}+\alpha_{t}z_{t}$, $f(h,r,t)$ can affect $g(h,r,t,z_{h},z_{r},z_{t})$.
>
> We do not mention the smoothing of $g(\cdot)$ because the choice of $g(\cdot)$ is flexible (not constraint). In principle, $g(\cdot)$ can be any function, we implement $g(\cdot)$ as the addition of $f(h,r,t)$ and $\alpha_{h}z_{h}+\alpha_{r}z_{r}+\alpha_{t}z_{t}$ for simplicity. And the smoothing condition is not a strong condition, common functions can satisfy it. Besides, the smoothing condition is a necessary condition for models optimized by stochastic gradient descent.
>
> Q5: The balancing of $\alpha_{h}$,$\alpha_{r}$ and $\alpha_{t}$.
>
> A5: $\alpha_{h}$, $\alpha_{r}$ and $\alpha_{t}$ are hyper-parameters to control the weight of confounders. The choices of them depend on the specific dataset because the data bias is different for different datasets as shown in Table 4. We choose them according to the result of MRR(-1,-1,-1) on the validation set.
>
> In Section 5.3, we conduct ablation studies to analyze which of the confounders $(Z_{H}, Z_{R}, Z_{T})$ influences the models most. The results show that $Z_{T}$ influences models most, $Z_{H}$ and $Z_{R}$ influence models little. The reason is that KGC is formulated as predicting the tail entities.
>
> We also have shown a sensitivity analysis of hyper-parameters $\alpha_{h}$,$\alpha_{r}$ and $\alpha_{t}$ in Appendix A Paragraph “Sensitivity Analysis”. The results show that our models are not sensitive to the hyper-parameters $\alpha_{h}$,$\alpha_{r}$ and $\alpha_{t}$.

---

> ### Author Response · Authors · 2022-11-15
> **Rebuttal for Reviewer L8AG, Part Two**
>
> Q6: Is there any existing metric for better comparison?
>
> A6: Our defined metric $\text{Metric}(\beta_{h},\beta_{r},\beta_{t})$ is a generalization of existing metrics MRR and H@N, which helps us evaluate the causal KGC models without a randomized test set. As far as we know, $\text{Metric}(\beta_{h},\beta_{r},\beta_{t})$ is the first metric to consider the popularity of entities and relations.
>
> Q7: The performance of Causal-xxx-1 and Causal-xxx-2 on FB15k-237 dataset.
>
> A7: Since the expressiveness of DistMult is $\textbf{not}$ enough, we need learnable confounders to enable the expressiveness. Thus, Causal-DistMult-2 is better than Causal-DistMult-1. Since the expressiveness of ComplEx is enough, ComplEx with learnable confounders suffers from overfitting. Thus, Causal-ComplEx-1 is better than Causal- ComplEx -2.
>
> The pair of (TransE,RotatE) is similar to the pair of (DistMult,ComplEx). Since the expressiveness of TransE is $\textbf{not}$ enough, we need learnable confounders to enable the expressiveness. Thus, Causal-TransE-2 is better than Causal-TransE-1 (on MRR and H@1 metrices). Since the expressiveness of RotatE is enough, RotatE with learnable confounders suffers from overfitting. Thus, Causal-RotatE-1 is better than Causal-RotatE-2.
>
> Q8: It would be better to use a different note for the head entity and tail entity.
>
> A8: We do $\textbf{not}$ use the same notation for the head entity and tail entity. We use the notation “h” for the head entity and the notation “t” for the tail entity.
>
> Q9: The authors might give a more advanced structure in the existing literature.
>
> A9: The causal graph in Figure 1(a) includes a large class of KGC models, such as translational-based KGC models and multiplicative KGC models and neural networks KGC models (except graph neural networks KGC models) [1]. The scoring functions of these types of KGC models only depend on a head entity h, a relation r and a tail entity t. Thus, the causal graph in Figure 1(a) is general enough. We will consider other types of causal graphs in the future work.
>
> [1] Jing Zhang, Bo Chen, Lingxi Zhang, Xirui Ke, and Haipeng Ding. Neural, symbolic and neural-symbolic reasoning on knowledge graphs. AI Open, 2:14–35, 2021a.
>
> Q10: Related work about explainable and interpretable knowledge graph methods.
>
> A10: Thank you for your suggestions. We have added the related work about explainable and interpretable knowledge graph methods in the rebuttal revision.

---

> > ### Comment · Reviewer_L8AG · 2022-12-04
> > **Response to Author**
> >
> > I thank the authors' efforts for making the responses, and the current version of the paper is relative more well-organized compared to the initial draft. However, I believe the writing and presentation of the paper still not be adequate. For example, writing the multiple lines of the mathematical formulation in the main-text, e.g., the equation derivation at the bottom of the page 5 (there are many other typos and writing issues in the paper I have not indicated here). In addition to the paper quality, my main concern is the inadequate contribution of the work to the community: both knowledge graph completion and causal knowledge are well-investigated in the existing literature, but some of key works are not mentioned in the paper or used for empirical comparisons. Therefore, I decide to keep my score and cannot support the publication of this work in the current stage.

---

### Official Review · Reviewer_YGuw · 2022-10-24

**Confidence:** 4
**Correctness:** 3
**Technical Novelty And Significance:** 3
**Empirical Novelty And Significance:** 2
**Recommendation:** 6

**Clarity, Quality, Novelty And Reproducibility:**

Clarity is high, although a few grammar and language errors interfere at certain places (the use of "Roughly" is inconsistent with current use, in particular).

Quality is moderate as there are three datasets considered (good), but the results are evaluated with a single metric, and thus no "sensitivity analysis" is carried out (poor); there is also a throwaway reference to group theory which does not seem to advance the paper in any way.

Novelty is moderate insofar as similar methods have been previously proposed and some of them (especially IPS) perform comparably well.

Reproducibility is low as no source code or link to a repository containing it is provided (although the methods are described fairly clearly, this is insufficient for reproducing the results as the learned parameters are missing, as is the code and the RNG seeds used to generate them).

**Strength And Weaknesses:**

Strengths: The overall presentation is clear, and the method is explained in a logical way that is easy to follow. The abstract sets the stage, the introduction reviews relevant work, and the methods clarify the novelty of the work. The results are presented fairly. The conclusion points out that much future work remains.

Weaknesses: These are easier to list for specific sections, which I have done below.

Abstract: "our causal KGC models achieve better performance than traditional KGC models" - on what inputs?

Intro: "the world is driven by causality rather than correlation" - over-generalisation, especially as causality is often more difficult to observe

Background: the embedding isn't motivated or explained; is it externally specified? learned in the graph construction process? In the tensor representation of the causal graph each entity and relation is an element of a set, not a vector, so this needs a clear explanation/illustration.

Methods:

1) While the entire process is motivated by causal inference and the removal of confounders, the simplifying assumptions ultimately amount to a. a frequency penalty and b. a learned entity-specific or relationship-specific penalty. Would it not have been easier to present the entire approach as a penalty-based one, and then provide the motivating causal inference-based derivation in the appendix?

2) The group-theoretic view does not provide any additional value or insight to the paper; I say this despite being a big fan of group theory myself. I recommend removing this section altogether.

3) A clearer differentiation from inverse propensity scoring should be provided by explicitly showing how IPS does something different from the frequency penalty introduced in the method.

Results: "different evaluation metrics" - it seems that you only used a single evaluation metric in the end, namely, Metric(-1,-1,-1)?

Discussion: "our model is [...] model-agnostic" is confusing; perhaps "our approach is [...]" instead?

Appendix: the standard deviation of frequency is not a great way to quantify bias; perhaps fitting a modified Zipf's law or computing a Gini coefficient could be more useful?

**Summary Of The Paper:**

The paper proposes a novel approach to the task of knowledge graph completion, which draws on insights from the field of causal reasoning to obtain an adjustment that should - in principle - help with bias correction in the data underlying the knowledge graph. It considers two types of artificial confounders, one based on the log-popularity and another one, based on a one-layer neural network (aka a perceptron).

**Summary Of The Review:**

Overall, a decent paper with a moderate advance over the current state-of-the-art; it is a "one-trick pony" paper, with a single idea being presented and implemented, and evaluated with a single metric.

---

> ### Author Response · Authors · 2022-11-15
> **Rebuttal for Reviewer YGuw**
>
> Thank you for your careful and detailed comments. We have polished the paper for a clearer statement.
>
> Q1: Abstract: "our causal KGC models achieve better performance than traditional KGC models"-on what inputs?
>
> A1: We have modified it to “Our causal KGC models achieve better performance than traditional KGC models on three benchmark datasets.”.
>
> Q2: Introduction: "the world is driven by causality rather than correlation"–over-generalization.
>
> A2: We have modified it to “Since correlation is not as reliable as causation”.
>
> Q3: Background: the notations
>
> A3: We slightly abuse the notation $f(\cdot)$, we use $f(\cdot)$ to denote both $f(h,r,t)$ and $f(\bf{h},\bf{r},\bf{t})$. In the rebuttal revision, we only use $f(\cdot)$ to denote $f(h,r,t)$ to avoid the misunderstanding. We may not fully understand your question. Is our explanation right?
>
> Q4: A view of penalty-based
>
> A4: We agree that the view of penalty-based can make our method easier. Thank you for your suggestions.
>
> Q5: The advantages and potential applications of the view of group theory.
>
> A5: The advantages of the view of group theory to the view of causal are in three folds. First, some relationships between variables are hard to describe by causal graphs. For example, for the ideal gas law $PV=nRT$, it is hard to say what causes what. In group theory, we use the invariance to represent the relationships between variables, which is easy to describe. Second, there is no metric to measure the relationships between causal graphs. In group theory, we use the notations of group theory to measure the relationships between groups, such as the order of groups, normal subgroups etc., which help us understand the invariance. Third, backdoor adjustment formula is only suitable for probability function. In group theory, we can use the group to act on any function. Thus, the view of group theory generalizes backdoor adjustment formula to any function.
>
> The potential applications of the view of group theory are in three folds. First, the view of group theory enables KGC models to solve out-of-distribution problem. In this paper, we learn a $G$ invariance scoring function from the biased data and apply a group action to the $G$ invariance scoring function to make it able to evaluate on unbiased data. Thus, the task of this paper can be seen as an out-of-distribution task. We can apply different group actions to the $G$ invariance scoring function to make it adapt to different test sets. Second, learning $G$ invariance scoring function directly may be difficult, the direct product decomposition and subgroup series allow us to simplify the group $G$. Third, the $G$ invariance is pre-specified in this paper. It is worth exploring how to learn the invariance from data automatically? Given a group action on $\mathcal{X}$, the orbits form a partition of $\mathcal{X}$. Thus, we can transform learning the invariance into learning the orbits of $\mathcal{X}$.
>
> We have rewritten Section 3.3 for a clearer statement.
>
> Q6: A clearer differentiation from inverse propensity scoring (IPS).
>
> A6: The IPS method reweights the weight of loss function for a batch of data, the weight is relevant to the popularity of entities and relations. Our method is to add an extra term to the scoring function. Thus, IPS method focuses on the loss function, our method focuses on the scoring function. We have added a more detailed statement of IPS in Appendix A Paragraph “Inverse Propensity Scoring”.
>
> Q7: Results: "different evaluation metrics" - it seems that you only used a single evaluation metric in the end, namely, Metric(-1,-1,-1)?
>
> A7: The “different” here means that we use three different evaluation metrices, MRR, H@1 and H@10. We use Metric(-1,-1,-1) to denote MRR(-1,-1,-1) or H@N(-1,-1,-1) as shown in Section 3.2. We have made a clearer statement.
>
> Metric(-1,-1,-1) can be seen as a combination of three metrics, Metric(-1,0,0), Metric(-1,0,0) and Metric(-1,0,0), which simultaneously takes into account the popularity of head entities, relations and tail entities, as shown in Section 3.2.
>
> Q8: Discussion: "our model is ... model-agnostic" is confusing.
>
> A8: Thank you for your suggestions.
>
> Q9: Appendix: Gini coefficients.
>
> A9: Thank you for your suggestions. We have added the results of Gini coefficients in Table 4 of the rebuttal revision, which is more accurate.
>
> Q10: Quality: no sensitivity analysis.
>
> A10: We have added the sensitivity analysis with respect to the hyper-parameters $(\alpha_{h},\alpha_{r},\alpha_{t})$ in Appendix A Paragraph “Sensitivity Analysis”. The results show that our models are not sensitive to the hyper-parameters $(\alpha_{h},\alpha_{r},\alpha_{t})$.
>
> Q11: Reproducibility: no source code.
>
> A11: We have submitted the source code in the supplementary material of the original version. Is there something wrong with the OpenReview website? We add an additional file about the settings of hyper-parameters in the supplementary material of the rebuttal revision.

---

### Official Review · Reviewer_v6Kd · 2022-10-31

**Confidence:** 3
**Correctness:** 3
**Technical Novelty And Significance:** 2
**Empirical Novelty And Significance:** 2
**Recommendation:** 5

**Clarity, Quality, Novelty And Reproducibility:**

This paper is poorly written, bears with various typos and incomplete sentences, and the technical details are hard to follow. Overall, I suggest the authors to rewrite the Introduction and Background sections completely, and try not to use repeated sentences in Abstract, Introduction and Conclusion.

**Strength And Weaknesses:**

Strength: It's an interesting ideal of trying to incorporate causal inference techniques into traditional KGC models, and the group theory perspective of KGC is quite novel.

Weakness: 1.I agree that in some sense, data bias can be treated as confounders in the causal model, however, I do not think it's appropriate to simply define the confounders as the "artificially designed confounders" and "learnable confounders" in this paper. Confounders are the type of variables that have causal impact over both treatment and outcome variables, but in the definition of "artificially designed confounders" in this paper, they are essentially treated as the variables that are impacted by treatment variables. In other words, the directions of certain arcs in the causal graph changed.
2. I appreciate the attempt of explaining KGC from the group theory perspective. However, I cannot see if there's an potential application or new insights from such explanation.

Other comments: 1. Too much identical sentences in Abstract, Introduction and Conclusion.
2. In multiple places, you wrote "In this paper, we propose causal KGC models to alleviate the issues ..." Here you should be more specific by saying "data bias issues".
3. You mentioned "causal graph" is multiple places of this paper. However, in the causal KGC model, all causal graphs are essentially identical and they have the same topology as the graph (b) in Figure 1. The reason why causal graph in traditional causal inference is important is because, it gives people a direct way of viewing the relationship between variables, which can help people identify what are confounders, what are instrumental variables etc. By introduction causal model into KGC, here different triplets (h,r,t) will still have no connecting arcs between them, except the newly added confounder variables $(Z_h, Z_r, Z_t)$. For that reason, I do not think you should highlight the concept of "causal graph" in this paper at all.
4. You mentioned a few times that "the main feature of causal is invariance and symmetry (Arjovsky et al., 2020)". However, I did not see that from your cited literature. Why is the main feature of causal is symmetry? A causes B means something totally different from B causes A.
5. First sentence in Section 2.1, "let $\epsilon$ denote(s) and ... denote(s)".
6. Bottom of page 4: "$P(y \mid h, \ldots, z_t)$ evaluates..." this is an incomplete sentence.
7. Bottom of page 4: "confounders (is)" should be "are".
8. When constructing the "learnable confounders", how to learn those neural networks?
9. On page 5, $z_t'$ should be $z_{t'}$.
10. Please add references to those ranking metrics $MRR$ and $H@N$.
11. Should not use the word $Metric$ as the name of your evaluation metric.
12. In Section 3.3, "should invariant" -> "should be an invariant".
13. In Section 3.3, "which may (do) not match ..."
14. Last sentence of Section 5.2: I cannot see the reason from the degree of data bias. Please explain more.

**Summary Of The Paper:**

This paper proposed a causal KGC model to alleviate the data bias issues that come from the data collection process, and further proposed an evaluation method to evaluate the causal KGC models on observation datasets. Moreover, the authors also provided a different perspective of viewing KGC from group theory. Numerical experiments showcased that the causal KGC models outperform traditional KGC models in most cases.

**Summary Of The Review:**

Based on my comments in the previous sections, even though this paper has certain merit of introducing causal model into KGC, but overall I do not think the contribution is significant enough to make this paper be accepted as the ICLR conference proceeding.

---

> ### Author Response · Authors · 2022-11-15
> **Rebuttal for Reviewer v6Kd, Part One**
>
> Thank you for your careful and detailed comments. We have polished the paper for a clearer statement.
>
> Q1: The directions of the arcs (from $Z_{H}/Z_{R}/Z_{T}$ to $H/R/T$) in the causal graph are changed.
>
> A1: The directions of the arcs in the causal graph are $\textbf{not}$ changed. We compute the value of the confounder $z_{h}/z_{r}/z_{t}$ from the observation data $X_{h,:,:}/X_{:,r,:}/ X_{:,:,t}$, this does $\textbf{not}$ mean that $H/R/T$ causes $Z_{H}/ Z_{R}/ Z_{T}$. For example, we can compute the rate of drowning deaths from ice cream sales, this does $\textbf{not}$ mean that ice cream consumption causes drowning. Thus, the functional relationships do not imply causal relationships.
>
> We can only identify the causal relationships between variables from the data generation process, which is described by the causal graphs in Figure 1. We have illustrated the causal relationships between variables (i.e., the edges in the causal graphs) in Section 3.1. The confounders $(Z_{H}/Z_{R}/Z_{T})$ can affect the data generation process, which results the generation of $\textbf{biased}$ observation data.
>
> Now the $\textbf{biased}$ observation data $X$ is given, we need to estimate the conditional probability $P(y|h,r,t,z_{h},z{r},z_{t})$ as shown in the backdoor adjustment formula. $P(y|h,r,t,z_{h},z_{r},z_{t})$ describes the correlation between $y$ and $(h,r,t,z_{h},z_{r},z_{t})$ $\textbf{rather than causation}$. To estimate $P(y|h,r,t,z_{h},z_{r},z_{t})$, we need to get the value of the confounders $(z_{h},z_{r}/z_{t})$. The confounders $(z_{h},z_{r}/z_{t})$ are reflected in the $\textbf{biased}$ observation data $X$, i.e., the confounder $z_{h}/z_{r}/z_{t}$ is a function of $X_{h,:,:}/X_{:,r,:}/ X_{:,:,t}$.
>
> Q2: The advantages and potential applications of the view of group theory.
>
> A2: The advantages of the view of group theory to the view of causal are in three folds. First, some relationships between variables are hard to describe by causal graphs. For example, for the ideal gas law $PV=nRT$, it is hard to say what causes what. In group theory, we use the invariance to represent the relationships between variables, which is easy to describe. Second, there is no metric to measure the relationships between causal graphs. In group theory, we use the notations of group theory to measure the relationships between groups, such as the order of groups, normal subgroups etc., which help us understand the invariance. Third, backdoor adjustment formula is only suitable for probability function. In group theory, we can use the group to act on any function. Thus, the view of group theory generalizes backdoor adjustment formula to any function.
>
> The potential applications of the view of group theory are in three folds. First, the view of group theory enables KGC models to solve out-of-distribution problem. In this paper, we learn a $G$ invariance scoring function from the biased data and apply a group action to the $G$ invariance scoring function to make it able to evaluate on unbiased data. Thus, the task of this paper can be seen as an out-of-distribution task. We can apply different group actions to the $G$ invariance scoring function to make it adapt to different test sets. Second, learning $G$ invariance scoring function directly may be difficult, the direct product decomposition and subgroup series allow us to simplify the group $G$. Third, the $G$ invariance is pre-specified in this paper. It is worth exploring how to learn the invariance from data automatically? Given a group action on $\mathcal{X}$, the orbits form a partition of $\mathcal{X}$. Thus, we can transform learning the invariance into learning the orbits of $\mathcal{X}$.
>
> We have rewritten Section 3.3 for a clearer statement.
>
> Q3: Too much identical sentences.
>
> A3: We have rewritten the paper to avoid identical sentences.
>
> Q4: You should be more specific by saying "data bias issues".
>
> A4: Thank you for your suggestions.

---

> ### Author Response · Authors · 2022-11-15
> **Rebuttal for Reviewer v6Kd, Part Two**
>
> Q5: The necessity of causal graphs, the generality of causal graphs in Figure 1 and the relationships between different triplets.
>
> A5: The causal graph is necessary, which helps us know the data generation process and finds the root of data bias. The root of data bias is that some variables (confounders) can influence the data generation process, while they are ignored by traditional KGC models. Thus, traditional KGC models are biased toward popular entities/relations. Therefore, the newly added variables $\{z_{H},Z_{R},Z_{T}\}$ are vital.
>
> The causal graph in Figure 1(a)/Figure 1(b) includes a large class of KGC models, such as translational-based KGC models, multiplicative KGC models and neural networks KGC models (except graph neural networks KGC models) [1]. The scoring functions of these types of KGC models only depend on a head entity h, a relation r and a tail entity t. Thus, the causal graph in Figure 1(a)/Figure 1(b) is general enough. We will consider other types of causal graphs in the future work.
>
> Our designed confounders can establish the relationships between different triplets. For example, if the artificially designed confounder $z_{h}$ is equal to $d$, then the number of non-zero entries of $X_{h,:,:}$ is equal to $d$, i.e., the number of the known true triplets with the head entity h is constraint to $d$. Thus, the confounder $Z_{H}$ establishes the relationships between triplets with the same head entity h.
>
> [1] Jing Zhang, Bo Chen, Lingxi Zhang, Xirui Ke, and Haipeng Ding. Neural, symbolic and neural-symbolic reasoning on knowledge graphs. AI Open, 2:14–35, 2021a.
>
> Q6: Why is the main feature of causal is invariance or symmetry?
>
> A6: To predict the outcome of an intervention, we rely on the properties assumed invariant after the intervention. For structural causal models framework, the do-calculus tells which conditionals remain invariant after an intervention. For potential outcome framework, the ignorability plays the same role.
>
> If A is the cause of B, then the relationship “A cause B” are invariant across experimental conditions, which will sponsor valid predictions across experimental conditions. Please see Section 4.3 of (Arjovsky et al., 2020) for more details.
>
> Q7: The typos in first sentence of Section 2.1, bottom of page 4, page 5 and Section 3.3.
>
> A7: We have modified them in the rebuttal revision.
>
> Q8: When constructing the "learnable confounders", how to learn those neural networks?
>
> A8: The neural networks are learned by optimizing the loss function in Page 5. The loss function is related to $g(h,r,t,z_{h},z_{r},z_{t})$, $g(h,r,t,z_{h},z_{r},z_{t})$ is related to $(z_{h},z_{r},z_{t})$ and $(z_{h},z_{r},z_{t})$ is the output of the neural networks. Thus, we can learn the neural networks end-to-end.
>
> Q9: Please add references to those ranking metrics.
>
> A9: We have added the references in the rebuttal revision.
>
> Q10: Should not use the word “Metric” as the name of your evaluation metric.
>
> A10: We have modified the word “Metric” to “W-Metric” (weighted metric).
>
> Q11: Please explain more of the last sentence of Section 5.2.
>
> A11: Thanks for the suggestions of Reviewer YGuw. We use the Gini coefficient of the popularity (which is a more accurate metric) instead of the standard deviation of the popularity to measure the data bias in the rebuttal revision. Larger Gini coefficients mean larger data bias. The Gini coefficients show that the degree of data bias on WN18RR dataset is less than that on FB15k 237 dataset or YAGO3-10 dataset. Thus, the improvement on WN18RR dataset is little. Please see Table 4 for more details.

---

### Decision · Program_Chairs · 2023-01-20

**Decision:**

Reject

**Justification For Why Not Higher Score:**

It is not ready for publication. Neither the importance of the contributions are clear not the writing of the paper is suitable for a good venue.

**Justification For Why Not Lower Score:**

N/A

**Metareview: Summary, Strengths And Weaknesses:**

The authors propose a causal knowledge graph completion (KGC) model to alleviate issues pertaining to mere correlation-based KGCs.
The reviewers felt that the paper’s writing and presentation of the material must be improved. They also had concerns about the significance of the work as for instance how it differs from other methods such as IPS. Perhaps a rewrite of the paper in light of reviewers concerns could improve and clarify the significance of the results.